global warming; ocean acidification; ocean deoxygenation; sea-level rise; detection and attribution

**Corresponding author:**
David S. Schoeman;
Email: dschoema@usc.edu.au

# Quantifying the ecological consequences of climate change in coastal ecosystems

David S. Schoeman[1,2] , Jessica A. Bolin[1] and Sarah R. Cooley[3]

[1]Ocean Futures Research Cluster, School of Science, Technology and Engineering, University of the Sunshine Coast, Maroochydore, Queensland, Australia; [2]Centre for African Conservation Ecology, Department of Zoology, Nelson Mandela University, Gqeberha, South Africa and [3]Ocean Conservancy, Washington, DC, USA

## Abstract

Few coastal ecosystems remain untouched by direct human activities, and none are unimpacted by anthropogenic climate change. These drivers interact with and exacerbate each other in complex ways, yielding a mosaic of ecological consequences that range from adaptive responses, such as geographic range shifts and changes in phenology, to severe impacts, such as mass mortalities, ecological regime shifts and loss of biodiversity. Identifying the role of climate change in these phenomena requires corroborating evidence from multiple lines of evidence, including laboratory experiments, field observations, numerical models and palaeorecords. Yet few studies can confidently quantify the magnitude of the effect attributable solely to climate change, because climate change seldom acts alone in coastal ecosystems. Projections of future risk are further complicated by scenario uncertainty – that is, our lack of knowledge about the degree to which humanity will mitigate greenhouse-gas emissions, or will make changes to the other ways we impact coastal ecosystems. Irrespective, ocean warming would be impossible to reverse before the end of the century, and sea levels are likely to continue to rise for centuries and remain elevated for millennia. Therefore, future risks to coastal ecosystems from climate change are projected to mirror the impacts already observed, with severity escalating with cumulative emissions. Promising avenues for progress beyond such qualitative assessments include collaborative modelling initiatives, such as model intercomparison projects, and the use of a broader range of knowledge systems. But we can reduce risks to coastal ecosystems by rapidly reducing emissions of greenhouse gases, by restoring damaged habitats, by regulating non-climate stressors using climate-smart conservation actions, and by implementing inclusive coastal-zone management approaches, especially those involving nature-based solutions.

## Impact statement

Human society deeply values coasts and the ecosystem services they provide. But navigating the challenge of coastal management over the coming decades to sustain these connections depends on an ability to identify and quantify the ecological consequences of climate change. Aiding in this task has been a sequence of Assessment Reports by the Intergovernmental Panel on Climate Change (IPCC), the most recent of which is its Sixth Assessment Report. Both this IPCC Report and associated studies emphasise that human-caused climate change has driven our oceans into states unprecedented over millennia, and that these changes have led to fundamental ecological impacts across all coastal ecosystems. These impacts exacerbate and are exacerbated by other human-caused impacts in the coastal zone. Projections of future risk mirror the impacts already observed, but they escalate with cumulative greenhouse-gas emissions. Although these conclusions are supported by multiple lines of evidence, progress beyond qualitative assessments is hampered by our inability to confidently disentangle the effects of interacting drivers of change. Difficulties in this regard escalate as the number of drivers considered increases. Promising avenues for progress include emerging collaborative initiatives, such as model intercomparison projects, and the more inclusive use of multiple knowledge systems. In the interim, however, reducing risks over the remainder of this century depends on rapidly reducing emissions, restoring damaged habitats, designing and deploying climate-smart conservation actions that alleviate non-climate stressors, and carefully managing existing and future coastal development, with an emphasis on nature-based solutions.

## Introduction

The world's coasts hold special places in human history and culture: settlements and cities have sprung up close to the sea because of the rich resources the ocean and coastal ecosystems provide, the transport and trade they facilitate, and the sense of place they instil (Neumann et al., 2015; Cooley et al., 2022). As a direct result, these coastal ecosystems face multiple escalating threats from

humanity, most of which exacerbate or are exacerbated by climate change. Navigating the challenge of coastal management over the coming decades therefore relies heavily on being able to identify and quantify the ecological consequences of climate change. Aiding in this task has been a sequence of Assessment Reports by the Inter-governmental Panel on Climate Change (IPCC), the most recent of which is its Sixth Assessment Report (IPCC, 2021, 2022). Here, we explore the main findings of that work, providing illustrative examples of climate-driven impacts and risks, identifying key challenges to progress, and briefly discussing promising avenues that might lead to the development of more robust, quantitative projections of future risk to coastal ecosystems due to climate change.

## Human-induced climate change has vastly altered the environmental conditions within which coastal ecosystems operate

Anthropogenic climate change has driven the physical and chemical conditions of coastal ecosystems (Table 1) to states that are unprecedented over millennia (Cooley et al., 2022). Approximately 93% of the excess incoming solar energy trapped by greenhouse gases is absorbed by the ocean. This has added 350 ZJ (1 ZJ = $10^{21}$ J) to the heat content of the oceans between 1958 and 2019, with an annual acceleration over the past decade. (Cheng et al., 2022). This has driven the heat content of the upper ocean to reach record levels in 2022, exceeding the previous record (2021) by approximately 10 ZJ (Cheng et al., 2023). Associated warming has been almost ubiquitous along coastlines (Lima and Wethey, 2012) and within estuaries (Scanes et al., 2020). The exception has been upwelling cells, where the increased prevalence of upwelling-favourable winds has resulted in local cooling, especially at higher latitudes (Bograd et al., 2023). Superimposed on this long-term trend of warming has been a rapid surge in the localised occurrence of anomalously warm waters that persists for days to many months, known as marine heatwaves (MHWs) (Hobday et al., 2016; Laufkötter et al., 2020; Sen Gupta et al., 2020). The frequency of MHWs has at least doubled since the 1980s, and MHW intensity has increased rapidly with ocean warming, as has the proportion of time the global ocean is subject to MHW conditions (IPCC, 2021).

**Table 1.** Estimates of magnitudes of observed and projected changes in climate-induced drivers pertinent to coastal ecosystems, as assessed by the IPCC. These estimates are global averages and it should be noted that in each case, considerable spatial variability is anticipated, especially in coastal areas. Unless otherwise stated, ranges in brackets represent 90% confidence intervals.

| Climate impact-driver | Observed change | Projected change |
| --- | --- | --- |
| Ocean warming | Considering the decade 2011–2020, the global ocean surface waters have warmed on average by 0.88°C (0.68–1.01°C) compared with the period 1850–1900, with 0.60°C (0.44–0.74°C) of this warming having occurred since 1980 (Fox-Kemper et al., 2021). | Relative to the 20-year period ending in 2014, global ocean surface temperatures in the last two decades of the century are projected to have warmed by 0.86°C (0.43–1.47°C) under SSP1-2.6, 1.51°C (1.02–2.19°C) under SSP2-4.5, 2.19°C (1.56–3.30°C) under SSP3-7.0, and 2.89°C (2.01–4.07°C) under SSP5-8.5 (Fox-Kemper et al., 2021). |
| Marine heatwaves | Over the course of the 20th century, marine heatwaves became more frequent and intense, with a rapid escalation in the 21st century: since the 1980s, the frequency of marine heatwaves has doubled, and their intensity and duration have rapidly increased (Fox-Kemper et al., 2021). | Relative to the 20-year period ending in 2014, marine heatwaves are projected to be four times more frequent in the last two decades of the century under SSP1-2.6, and eight times more likely under SSP5-8.5 (Fox-Kemper et al., 2021). |
| Stratification and deoxygenation | Stratification of the upper 200 m of the ocean water column has increased by approximately 5% since the 1970s (Arias et al., 2021). Over the same period, the subsurface ocean (100–600 m depth) lost 2% of its total dissolved oxygen, resulting in the identification of >700 hypoxic (<2 mg $O_2$ $L^{-1}$) coastal regions (Canadell et al., 2021). | Over the course of the 21st century, the ocean water column will continue to stratify, and subsurface waters are projected to transition to historically unprecedented conditions, with dissolved oxygen in the last two decades of the century declining by between 6.4 ± 2.9 mmol $m^{-3}$ (under SSP1-2.6) and 13.3 ± 5.3 mmol $m^{-3}$ (under SSP5-8.5), relative to the period 1870–1899 (Canadell et al., 2021). |
| Acidification | The pH of surface waters in the open ocean has declined by 0.012–0.104 pH units since the 1970s, and acidification of deeper waters has become ubiquitous since the 1980s (Gulev et al., 2021). | Relative to the period 1870–1899, ocean surface pH in the last two decades of the 21st century is projected to have declined by 0.16–0.44 pH units under SSP1-2.6 and SSP5-8.5, respectively (Canadell et al., 2021). |
| Sea-level rise | Over the period 1901–2018, global mean sea level rose by 201.9 mm (150.3–253.5 mm), with 44.3 mm (38.6–50.0 mm) of this rise since 2006 at a rate of 3.7 mm $yr^{-1}$ (3.2–4.2 mm $yr^{-1}$) (Fox-Kemper et al., 2021; Gulev et al., 2021). High-tide flooding events that occurred five times per year during the period 1960–1980 occurred, on average, more than eight times per year during the period 1995–2014 (Fox-Kemper et al., 2021). | Ignoring high-impact–low-likelihood outcomes, such as Antarctic marine ice-cliff instability, global mean sea levels (relative to those for the period 1995–2014) are projected to rise by between 190 mm (66% confidence: 160–250 mm) and 230 mm (200–290 mm) by 2050 under SSP1-2.6 and SSP5-8.5, respectively. Corresponding projections for 2100 are 440 mm (320–620 mm) and 770 mm (630–1010 mm), respectively. Associated rates of sea-level rise are 4.8 mm $yr^{-1}$ (3.5–6.8 mm $yr^{-1}$) to 7.2 mm $yr^{-1}$ (5.6–9.7 mm $yr^{-1}$) over the 20-year period centred on 2050 under SSP1-2.6 and SSP5-8.5, respectively. Corresponding projections for the 20-year period centred on 2090 are 5.2 mm $yr^{-1}$ (3.2–8.0 mm $yr^{-1}$) and 12.1 mm $yr^{-1}$ (8.6–17.6 mm $yr^{-1}$), respectively (Fox-Kemper et al., 2021). These projections mean that historically extreme sea levels (i.e., 1-in-100-year events for the period 1995–2014) might occur annually (or more frequently) across 19–31% of locations by 2050, rising to 60–82% of locations by 2100 under SSP1-2.6 and SSP5-8.5, respectively (Fox-Kemper et al., 2021). |

As the oceans have warmed, the solubility of gases has decreased, resulting in a strong trend in declining dissolved oxygen content – a process known as ocean deoxygenation (Canadell et al., 2021). Ocean stratification is enhanced by warming, reducing mixing (ventilation), altering nutrient redistribution, and exacerbating deoxygenation, especially in subsurface waters. Oxygen minimum zones (OMZs) – areas of low oxygen concentration in the upper ocean that are especially apparent in tropical regions – have been expanding at least since 1960 (Zhou et al., 2022). An exception to warming-driven deoxygenation is found where intensifying upwelling brings cold, low-oxygen water to the surface (Canadell et al., 2021; Bograd et al., 2023). A further direct impact of warming is melting ice, both at sea and grounded on land. The latter – together with the thermal expansion of seawater – has contributed to accelerating rates of sea-level rise, now averaging >3 mm per year, faster than any time in at least the last 3,000 years (Fox-Kemper et al., 2021; Le Cozannet et al., 2022).

Increasing atmospheric $CO_2$ concentrations have resulted in decreasing pH of ocean waters – known as ocean acidification – so that surface-water pH is now unusually low in the context of the past 2 million years (Arias et al., 2021). Finally, changes in precipitation, stratification and ice-melt have enhanced contrasts in salinity between relatively salty and relatively fresh parts of the ocean (Cheng et al., 2020).

## Ocean conditions are projected to continue diverging from their pre-industrial state, with the magnitude of change depending on cumulative emissions

Most of the observed changes in ocean conditions due to anthropogenic climate change (Table 1) are irreversible on centennial to millennial scales, given present mitigation tools (IPCC, 2021). Warming of the upper ocean by 2100 is projected to range 2–8 times that experienced over the period 1971–2015, resulting in more frequent and intense MHWs, greater deoxygenation, increased stratification, faster ice-melt and accelerating sea-level rise. Ocean acidification is expected to intensify and salinity contrasts to be enhanced (Arias et al., 2021).

## Confirming that climate change has caused ecological responses

While change in Earth's climate system has been unambiguously attributed to anthropogenic greenhouse-gas emissions (IPCC, 2021), distinguishing the role of anthropogenic climate change in altering ecological systems from the roles of other potential drivers has proven more problematic and contentious (Brander et al., 2011; Pielke, 2011; Stocker et al., 2011). Such attribution has been especially difficult in coastal systems, where human impact is ubiquitous (Williams et al., 2022; Allan et al., 2023) and drivers of change compete with each other amidst naturally variable conditions, confounding unambiguous interpretations (Cooley et al., 2022; Friess et al., 2022). In such systems, attribution instead usually comprises a sequence of steps (Figure 1), often involving multiple lines of evidence (Parmesan et al., 2013; Hansen et al., 2016; Phillips, 2023).

The first step involves identifying a hazard. This could be a change in a climate variable that is known to be affected by anthropogenic greenhouse-gas emissions (i.e., a climate-induced driver), but it could equally be a phenomenon caused by a change in climate, including a management action taken to mitigate an

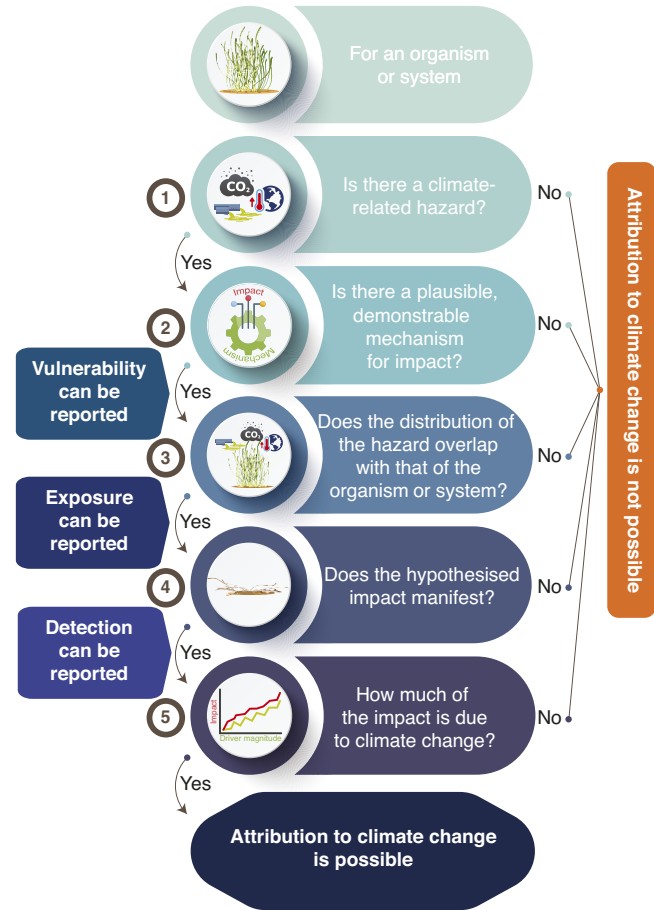

**Figure 1.** Steps involved in detecting and attributing an impact of climate change on an organism or ecological system. Note that the final step of attribution is seldom straightforward, instead often involving inference on the basis of multiple lines of evidence.

existing climate-change impact (e.g., the construction of a seawall, Simpson et al., 2021). The second requirement is to demonstrate that the organism or system is vulnerable to that hazard – that is, that there is a plausible and demonstrable mechanism for the putative effect. Third, the organism or system must experience – be exposed to – the hazard. Fourth, an exposed organism or system must demonstrate the anticipated change – a process known as "detection" of a climate impact. Ideally, detection is accompanied by an analysis that separates the putative climate response from responses to other non-climate drivers experienced by the organism or system (Hansen et al., 2016) – this constitutes formal "attribution" of a climate impact (Figure 1). Gonzalez et al. (2023) provide a more detailed discussion of detection and attribution and propose a quantitative framework for these processes as they apply to changes in biodiversity and other impacts to ecosystems (Parmesan et al., 2013; Ara Begum et al., 2022).

## Climate-induced drivers have greatly impacted life in the ocean and along its coasts

### Ocean warming

Temperature is a central driver of metabolic processes and therefore a key driver of ecological responses (Pörtner, 2021). Because the specific heat of seawater is around four times that of air, the

oceans have warmed only half as much as the atmosphere as a result of anthropogenic climate change, despite absorbing more than nine times the thermal energy (Fox-Kemper et al., 2021). But this property of seawater also means that ocean temperatures are generally less variable in space and time than those on land, resulting in marine organisms having narrower thermal tolerance ranges (Pinsky et al., 2019; Pörtner, 2021), and a greater predisposition towards occupying the full extent of these ranges than that of their terrestrial counterparts (Sunday et al., 2012). This renders marine biodiversity more vulnerable to warming than its terrestrial counterparts, causing a wide range of impacts. Among the more common consequences of the long-term trend in warming for coastal biodiversity are range shifts and tropicalisation, and changes in phenology.

### Range shifts, tropicalisation and depth shifts

Warming of the ocean surface since the 1950s has shifted marine taxa and communities poleward at an average (± 90% confidence interval) of 59.2 ± 15.5 km per decade (Lenoir et al., 2020; Fredston et al., 2021). Susceptibility to range shifts varies strongly by functional group, with short-lived, fast-growing planktonic organisms shifting their ranges much faster than longer-lived and sessile species, or species with fixed breeding sites (Poloczanska et al., 2013; Pinsky et al., 2020). Although range edges of coastal species in tropical to temperate waters generally maintain the species' thermal niches as the climate warms (Lenoir et al., 2020; Fredston et al., 2021), this tendency is both stronger and more common at the leading (cool) range edge than at the trailing (warm) range edge (Poloczanska et al., 2013; Fredston-Hermann et al., 2020; Pinsky et al., 2020). Pinsky et al. (2020) provide a detailed discussion of potential explanations for this phenomenon, including physiology, behaviour, evolution, dispersal and species interactions, but answers are elusive, and this question remains a topic of active research (e.g., Usui et al., 2023).

Nevertheless, arguably the most ubiquitous consequence of differential range shifts at leading and trailing range edges is the arrival of warm-affinity species in communities previously dominated by – and still occupied by – species of cooler provenance (Vergés et al., 2014; Chaudhary et al., 2021; Favoretto et al., 2022; Fujiwara et al., 2022). This process, known in low- to mid-latitudes as tropicalisation, in combination with the closely related process of deborealisation – the loss of cool-associated species from high-latitude places – results in the continual reassembly of biotic communities in coastal waters (McLean et al., 2021). When tropicalisation involves the arrival of herbivores, seagrasses and macroalgal habitat can be rapidly transformed (Vergés et al., 2016, 2022; Schuster et al., 2022; Santana-Garcon et al., 2023), reshaping entire ecosystems and their functioning (Peleg et al., 2020). But other arriving taxa can have equally profound impacts (de et al., 2022). Despite evidence that "healthy" ecosystems with relatively intact biodiversity can resist climate change of the magnitude already witnessed (Bates et al., 2014; Frid et al., 2023; Ziegler et al., 2023), the inevitability of range shifts identifies the network effects of tropicalisation and deborealisation as an urgent research priority.

Although range shifts are the most commonly studied ecological response to warming, some coastal species might respond to warming surface waters by seeking refuge at depth (Giraldo-Ospina et al., 2020). Evidence of such depth shifts, however, remains somewhat contradictory (Chaikin and Belmaker, 2023; Rubenstein et al., 2023).

### Marine heatwaves (MHWs)

MHWs can expose marine life to conditions beyond those projected for the end of this century for periods ranging from days to months (Sen Gupta et al., 2020; Koehlinger et al., 2023). It is therefore unsurprising that MHWs impact all levels of marine life, from the ecophysiology of individual organisms to the structure of marine communities (Smith et al., 2023). Although some effects can be beneficial, especially at high latitudes – for example, increased productivity or reproductive success in populations close to their leading range edge (Smith et al., 2019; Favoretto et al., 2022; Magel et al., 2022) – many effects are detrimental. MHWs can reduce breeding success (Hays et al., 2021; Rogers et al., 2021), cause trailing-edge extirpations and consequent regime shifts (Arafeh-Dalmau et al., 2019; Wernberg, 2021; Félix-Loaiza et al., 2022), facilitate geographic shifts at leading range edges (Smith et al., 2019; Favoretto et al., 2022; Coulson et al., 2023) and benefit non-native invasive species (Arafeh-Dalmau et al., 2019), drive mass mortalities in natural (Piatt et al., 2020) and aquaculture (Smith et al., 2021) settings, exacerbate infectious disease epidemics (Harvell et al., 2019; Claar and Wood, 2020; Genin et al., 2020), and impact habitat-forming taxa, including seagrasses (Strydom et al., 2020), kelps (Arafeh-Dalmau et al., 2019; Wernberg, 2021) and corals (Goreau and Hayes, 2021; Speare et al., 2022; van Woesik et al., 2022). However, the magnitude of effects varies by taxon, location, ecosystem type and health, and even genotype, as well as the intensity, duration, abruptness of onset and return interval of the MHW event (Smith et al., 2019; Fox et al., 2021; Suryan et al., 2021; Magel et al., 2022; Shlesinger and van Woesik, 2023; Ziegler et al., 2023).

### Phenology

Another conspicuous impact of ocean warming is the earlier attainment of typical spring temperatures and later attainment of autumn temperatures, which are both important in setting the timing of key seasonal events, such as breeding and migration. Although the timing of such seasonal events is not regulated by temperature signals alone (Ducklow et al., 2022; Whelan et al., 2022), a meta-analysis of phenological studies indicates that ocean warming has driven spring events 4.3 ± 1.8 days to 7.5 ± 1.5 days earlier per decade among planktonic organisms and 3.0 ± 2.1 days earlier per decade for fish (Cooley et al., 2022). Although there is more and stronger evidence for climate-driven phenological shifts among small, short-lived taxa (Cooley et al., 2022), recent evidence confirms such responses among large, long-lived taxa, including whales (Ganley et al., 2022; Pendleton et al., 2022; Shuert et al., 2022) and sharks (Hammerschlag et al., 2022). Moreover, since both range shifts and phenological shifts are responses to warming, it is unsurprising that taxa with high geographic fidelity (e.g., those with specific spawning or nesting requirements) might demonstrate stronger phenological responses than those that more readily shift ranges (Chust et al., 2023).

### Deoxygenation

Progressive loss of oxygen – deoxygenation – has been observed in the ocean interior since the mid-twentieth century (Canadell et al., 2021). Normally, oxygen enters the upper ocean from the atmosphere and from photosynthesis by aquatic vegetation (including phytoplankton), then vertical mixing moves oxygen into the deep ocean, where it is respired by heterotrophic marine organisms. About 15% of the observed deoxygenation is attributed to warming-induced decreases in oxygen solubility, and the rest is attributed to increased stratification (Canadell et al., 2021).

Deoxygenation is transforming marine communities by increasing individual species' migration, replacement and loss (Cooley et al., 2022) by, for example, altering the dynamics of aquatic infectious diseases (Burge and Hershberger, 2020; Byers, 2021) and threatening tropical shallow-water coral reefs with lethal and sublethal effects (Hughes et al., 2020; Pezner et al., 2023). In midwaters, deoxygenation is thought to compress habitat for pelagic oceanic fish species and temporarily increase catchability (Breitburg et al., 2018).

### Acidification

The decrease in surface ocean pH observed over the past 40 years due to the uptake of anthropogenically released atmospheric $CO_2$ has altered the water chemistry surrounding upper-ocean ecosystems more than in the past 26,000 or more years (Arias et al., 2021). Ocean acidification can have a variety of effects on biological processes: higher aquatic dissolved $CO_2$ concentrations tend to increase photosynthesis of some primary producers, while higher $H^+$ ion concentrations (i.e., greater acidity or lower pH) tend to challenge calcification – the biological creation of calcium carbonate shells and skeletons – for several animals or planktonic species, especially juveniles (Doney et al., 2020). Loss of juvenile Pacific oysters in aquaculture facilities (Barton et al., 2015) and increased bioerosion and dissolution of tropical corals in nature have been attributed to ocean acidification, but the complexity and variety of the effects of acidification on marine species, along with species' exposure to multiple simultaneous drivers, makes attributing many individual and most ecosystem-scale outcomes to ocean acidification extremely challenging (Doo et al., 2020).

### Sea-level rise

Thermal expansion of the ocean and, more recently, freshwater input from the loss of ice mass from terrestrial glaciers are driving up the global mean sea level (Fox-Kemper et al., 2021). Because coastlines can be subsiding (e.g., due to freshwater extraction) or experiencing isostatic rebound from the last glaciation (Durand et al., 2022), this sea-level rise is experienced as the change in the mean sea level relative to the land – relative sea-level rise. Almost all intertidal and shallow subtidal coastal ecosystems are sensitive to relative sea-level rise (Cooley et al., 2022). Observed impacts include flooding at high-tide extremes (Lawrence et al., 2022); salinisation of coastal soils, wetlands and the upper reaches of estuaries, with associated ecosystem transitions (Peteet et al., 2018; Andres et al., 2019; Kirwan and Gedan, 2019; Grieger et al., 2020; Eswar et al., 2021); increased erosion (e.g., Peteet et al., 2018); and coastal storm and flood damage (e.g., Strauss et al., 2021). Counterintuitively, relative sea-level rise can also result in accretion of intertidal sediments in areas where wetland vegetation can generate or trap sediments at rates exceeding those of relative sea-level rise (e.g., Marx et al., 2020; Saintilan et al., 2020).

Despite these observations, the impacts of relative sea-level rise are compounded with and confounded by other anthropogenic stressors at the coast, as well as the widespread deployment of countermeasures, including beach nourishment and other forms of coastal restoration and protection (Cooley et al., 2022). Global analyses of relatively coarse-scale imagery suggest the net effects of these processes have resulted in the loss of 15% of tidal flats since 1984 (Mentaschi et al., 2018; Murray et al., 2022), but with a corresponding number of the world's beaches accreting (28%) as eroding (24%) (Luijendijk et al., 2018).

### Ice loss

The effects of ice loss on coastal ecosystems are so far most keenly felt in the Arctic (Meredith et al., 2019; Cooley et al., 2022). Here, the formation, melting and persistence of sea ice drives seasonal patterns of coastal productivity, breeding and feeding opportunities, and connectivity (Le Moullec and Bender, 2022). Sea ice can be disruptive, through processes like benthic scouring, but can also be protective, through processes like buffering of coastal erosion (Lebrun et al., 2022). Irrespective, loss of ice in Arctic coastal systems can have cascading impacts (Meredith et al., 2019; Cooley et al., 2022), including the poleward movement of primary productivity driven by spring melt, with concomitant impacts for benthic and pelagic communities and the predators that feed on these (Brandt et al., 2023), including iconic species such as polar bears and walruses (Lebrun et al., 2022; Alabia et al., 2023; Kellner et al., 2023). Changes in ice phenology also impact phenology and breeding success among seabirds (Cusset et al., 2019; Descamps et al., 2019; Golubova, 2021). Despite these examples of impact, there is considerable variation among taxa and locations (Gutowsky et al., 2022; Grémillet and Descamps, 2023). Trends in ice loss and their attribution to climate change are both more uncertain in the Antarctic (Fox-Kemper et al., 2021; Cooley et al., 2022).

### Other climate-induced drivers

Coastal ecosystems and their resident organisms are variously sensitive to a range of climate-induced drivers beyond those discussed above (Cooley et al., 2022). Included amongst these are drivers, such as changes in ocean salinity and stratification, that are not yet expected to elicit ecological responses large enough to be detectable as climate-change impacts. Also included are changes in wave height and power (Young and Ribal, 2019; Odériz et al., 2021) and ocean circulation (e.g., Hu et al., 2020), which are yet to be confidently detected and attributed (Fox-Kemper et al., 2021; Gulev et al., 2021), and atmospheric phenomena that are difficult to predict, let alone project, such as tropical cyclones and storms. The latter can impact coastal systems such as vegetated wetlands and exposed sandy beaches, but with effects that are case-specific, and sometimes counter-intuitive. For example, the precipitation, wind and wave action associated with storms can rearrange coastal sediments, causing erosion in some places and accretion elsewhere (Xie et al., 2017; Armitage et al., 2020; Mo et al., 2020; Wang et al., 2020). In other cases, impacts can be indirect: heavy precipitation associated with storms can increase estuarine nutrient loads via runoff from adjacent land, causing or exacerbating eutrophication and stimulating HABs (Phlips et al., 2020; Dai et al., 2023), sometimes causing large-scale marine mammal, bird, and fish kills (Adams et al., 2019). Similarly, strong winds from tropical storms and cyclones can be beneficial for mangroves (Castañeda-Moya et al., 2020; Feher et al., 2020), or cause ephemeral damage (Armitage et al., 2020; Branoff, 2020), but they can also initiate regime shifts involving peat collapse and transition to mudflats (Chambers et al., 2019; Osland et al., 2020).

### Compound events

Although discussion so far has focused on the direct impacts caused by individual climate-induced drivers, in real-world situations, none of these operate in isolation. Instead, they combine and interact in various ways. This complicates the task of attributing

observed ecological responses to any single driver (Parmesan et al., 2013). For example, because ocean acidification and deoxygenation both depend on the solubility of gases in seawater – which is temperature dependent – these climate-induced drivers operate in concert with ocean warming to change the physiological suitability of coastal waters for marine fish and invertebrates.

Because temperature fundamentally affects the metabolism, motility, feeding efficiency and breeding success of marine organisms (Grady et al., 2019), ocean warming can also modify ecological interactions. Among the many examples of this phenomenon, one is of increasing concern: the host–pathogen interactions that drive outbreaks of infectious diseases in a range of coastal and marine taxa (Harvell et al., 2002; Randall and van Woesik, 2015; Cohen et al., 2018; Harvell et al., 2019). Since metabolic activity in ectothermic fish is temperature-dependent, warming temperatures, in concert with parasite exposure, are likely to facilitate the proliferation of disease-causing organisms and affect the health of fish hosts (Scharsack et al., 2021). In some cases, parasites grow faster and produce more viable eggs and offspring, which can lead to a rise in infection pressure, increased virulence, pathogenicity or expanded ranges for the parasite (Harvell et al., 2002; Arriaza et al., 2010; Cohen et al., 2018; Scharsack et al., 2021), and reduced fitness and/or mortality for the host. This can indirectly lead to trophic cascades in the warming habitat through changes to predation rates, thereby affecting ecosystem functioning (Harvell et al., 2019; Scharsack et al., 2021).

Such compounded combinations of hazards in marine systems result in rates of extirpation twice as high as those experienced by terrestrial taxa (Pinsky et al., 2019). On a global scale, the progressive loss of tropical biodiversity Chaudhary et al. (2021) provides stark evidence of these aggregate impacts of changes in the physical and chemical state of the ocean, in line with data from experiments and the palaeorecord (Reddin et al., 2020; Penn and Deutsch, 2022).

## The effects of climate change worsen and are worsened by the impacts on marine life of non-climate anthropogenic drivers

Not only do climate-induced drivers interact with each other, but their effects also modify and are modified by the effects of non-climate anthropogenic drivers (Sage, 2020; Gissi et al., 2021; Cooley et al., 2022). This is particularly true – but under-recognised – in coastal ecosystems, few of which remain untouched by human activities (Williams et al., 2022; Allan et al., 2023), rendering them especially vulnerable to the coupled climate and biodiversity crises (Pörtner et al., 2023).

The escalation of interactive effects of climate-induced drivers and other anthropogenic stressors is ubiquitous in coastal ecosystems (Halpern et al., 2019; He and Silliman, 2019; Gissi et al., 2021). Although examples abound (Table 2), we will restrict our brief discussion here to impacts on tropical coral reefs as a case study.

Corals are important habitat-forming species in tropical waters that support exceptionally high biodiversity (Fisher et al., 2015; Hughes et al., 2017) and provide extensive ecosystem services (Eddy et al., 2021). Yet many coral taxa are sensitive to climate change, especially through ocean warming and acidification, as well as to other anthropogenic stressors, such as nutrient and sediment loading (Hughes et al., 2017; Ellis et al., 2019; Cornwall et al., 2021; Zhao et al., 2021; Cooley et al., 2022). Importantly, both vulnerability to and recovery from the impacts of climate change are affected by local anthropogenic stressors (França et al., 2020; Cramer et al., 2021; Donovan et al., 2021). This is a double-edged sword: where reefs are exposed to both climate change and other human impacts, consequences can be severe; but this also means that well-designed climate-smart conservation interventions in these places, which both alleviate non-climate human impacts and deploy complementary strategies, should reduce vulnerability to climate change (Mellin et al., 2019; França et al., 2020; Dutra et al., 2021; Kuempel et al., 2022), at least in the short to medium term. Many other coastal ecosystems are less well-studied but would benefit equally from climate-smart conservation planning approaches (Brown et al., 2022; Doxa et al., 2022; Buenafe et al., 2023).

## Projecting future risks of climate change in coastal ecosystems

Given the difficulties in detecting and attributing climate-change impacts in coastal ecosystems, it should be no surprise that projecting future risks comes with even greater uncertainties. In some instances, projected risks of climate change are inferred from magnitudes of projected change in climate-driven hazards (Table 1, Figure 2), combined with the same established (or inferred) sensitivities to these drivers as are used in attributing observed impacts (Figure 1). In such cases, only the direction of change can be projected with any confidence (e.g., Hughes et al., 2020; Friess et al., 2022). In other cases, statistical models – including, but not limited to species distribution models – are used to map ecological responses against climate-driven hazards, and then this model is used to project the magnitude of future change in that response variable (e.g., Moltó et al., 2021; Van der Stocken et al., 2022; Chaudhary et al., 2023). More sophisticated, still, are ecosystem models that couple multiple environmental drivers to multiple interacting ecological response variables; these models are then used to extrapolate those interacting relationships forward under projected future climates (e.g., Moullec et al., 2019; Tittensor et al., 2021). All of these methods assume that the underlying models are transferable in time (and sometimes space), despite known problems with this assumption (Yates et al., 2018; Neupane et al., 2022; Rousseau and Betts, 2022). But as with the detection of climate impacts, confidence in their attribution – and therefore their utility as predictors of future responses – increases in the presence of multiple lines of evidence.

An alternative approach that does not rely on projecting established relationships forward in time, involves using palaeodata to estimate the magnitude of ecological responses to past climate states analogous to projected future climates (Fordham et al., 2020). Examples of such palaeodata for coastal ecosystems include reef and sediment cores (Jones et al., 2019; Cohen et al., 2020; Cramer et al., 2021; Hesterberg et al., 2022; Bograd et al., 2023). The advantages of palaeo-analogues of future climate are that they potentially account for natural adaptation in the taxa or systems impacted, and that human impacts are effectively eliminated (Kiessling et al., 2023). But questions about the transferability of estimates remain.

Beyond the approaches used to project future risks of climate change, it is important, also, to ensure that the future being assessed is plausible. Recent reviews (Burgess et al., 2023; Schoeman et al., 2023) provide detailed analysis of the use of future scenarios (described in Table 3), as used in climate-change

**Table 2.** Examples of interactions among climate-induced drivers and other anthropogenic stressors in coastal ecosystems.

| Non-climate driver | Interacting climate-induced drivers | Mechanism of impact | Ecosystems impacted | References |
|---|---|---|---|---|
| Runoff of fertilizers or organic matter | Warming, acidification, deoxygenation, sea-level rise | Nutrients released into the water column stimulate a pulse of primary production. When nutrients are depleted, secondary production consumes oxygen, leading to hypoxic or even anoxic, acidic conditions. | Estuaries, lagoons, deltas, shallow nearshore waters, including seagrass beds | Nelson and Zavaleta (2012); Brauko et al. (2020); DeCarlo et al. (2020); Wooldridge (2020); Dai et al. (2023) |
| Disturbance of organic-rich sediment | Warming, acidification, deoxygenation | | Estuaries, lagoons, deltas, shallow nearshore waters, including seagrass beds | Simone et al. (2021); Zhu et al. (2021); Smeaton and Austin (2022) |
| Coastal infrastructure | Sea-level rise | The mass of infrastructure and the abstraction of groundwater can lead to subsidence of coastal land, aggravating the effects of sea-level rise. | Shorelines of estuaries, lagoons, deltas | Rossi and Toran (2019); Befus et al. (2020); Bosserelle et al. (2022) |
| | | Infrastructure sets a hard limit to inland migration of coastal habitats in response to rising sea levels. This phenomenon is known as coastal squeeze. | Mangroves, saltmarshes, sandy beaches | Borchert et al. (2018); Lithgow et al. (2019) |
| Resource use | Warming | Fishing can impose additional sources of mortality on fish populations and benthic habitats, including seagrass beds. This can alter community structure and exacerbate the effects of ocean warming. | Estuaries, lagoons, nearshore waters | Brander (2007); Grech et al. (2012); Townhill et al. (2019) |
| | Warming, acidification, deoxygenation | By disturbing organic-rich sediment, some fishing methods, like bottom trawling, can exacerbate deoxygenation and acidification. | Estuaries, lagoons, deltas, shallow nearshore waters | De Leo et al. (2017); Bradshaw et al. (2021); Corell et al. (2023) |
| | Sea-level rise | Harvesting of trees for building material and fuel can make mangroves more susceptible to habitat transitions driven by sea-level rise. | Mangroves | Ward et al. (2016) |
| Impoundment of rivers | Sea-level rise | Reduction of freshwater input can accelerate upstream penetration of saline waters. | Estuaries | Herbert et al. (2015); Bricheno et al. (2021); Costa et al. (2023); Khondoker et al. (2023) |
| | | Reduced supplies of terrigenous sediments can exacerbate coastal erosion. | Sandy beaches | Tuck et al. (2021); Gao et al. (2023) |

ecology. Results reveal that the practice of focusing exclusively on RCP8.5/SSP5-8.5 – often designated a "high-emissions scenario" – is common. But even when complemented by RCP2.6/SSP1-2.6 – commonly designated a "low-emissions scenario" – the use of this extreme scenario is problematic for projections out to 2100: while SSP1-2.6 is still attainable over this timeframe, SSP5-8.5 is highly unlikely (Hausfather and Peters, 2020), despite being potentially useful over the near- to mid-term (Schwalm et al., 2020). Instead, SSP2-4.5 (Table 3) is believed to be the most plausible of the common long-term scenarios, arguably along with SSP1-2.6, and while SSP3-7.0 is a better "business-as-usual" scenario than SSP5-8.5, especially in the long term, it is also reasonably implausible in some regards (Burgess et al., 2023). It is important to note, however, that the plausibility of future emissions scenarios depends heavily on whether countries' ambitions to reduce such emissions will be matched with strong action (Rogelj et al., 2023). Irrespective, since much of the literature on projected impacts in the coastal zone by 2100 relies on RCP8.5/SSP5-8.5, significant caution is warranted when interpreting these projections.

## Projected future ocean conditions increase risks to ocean and coastal systems, including elevated risk of regional extirpations and global extinctions

A clear message from the IPCC Sixth Assessment Report cycle, including its Special Reports, is that risks from climate change escalate with the magnitude and duration of warming caused by greenhouse-gas emissions (Magnan et al., 2021; IPCC, 2023): every increment beyond 1.5°C of warming matters. Nowhere on Earth is this message more pertinent than along the world's coasts, where warming is faster than for the global ocean as a whole (Figure 2; Lima and Wethey, 2012; Varela et al., 2023). The thermal inertia of the ocean, together with the acknowledgement that sea levels will continue to rise for centuries and remain elevated for millennia (Fox-Kemper et al., 2021), means that there are few easy ways to reduce risks of climate change to coastal ecosystems over the remainder of this century. But rapidly reducing emissions, alleviating non-climate anthropogenic stressors, and attempting to restore damaged habitats will provide more operating room for the full adaptation toolkit. This includes revising institutions

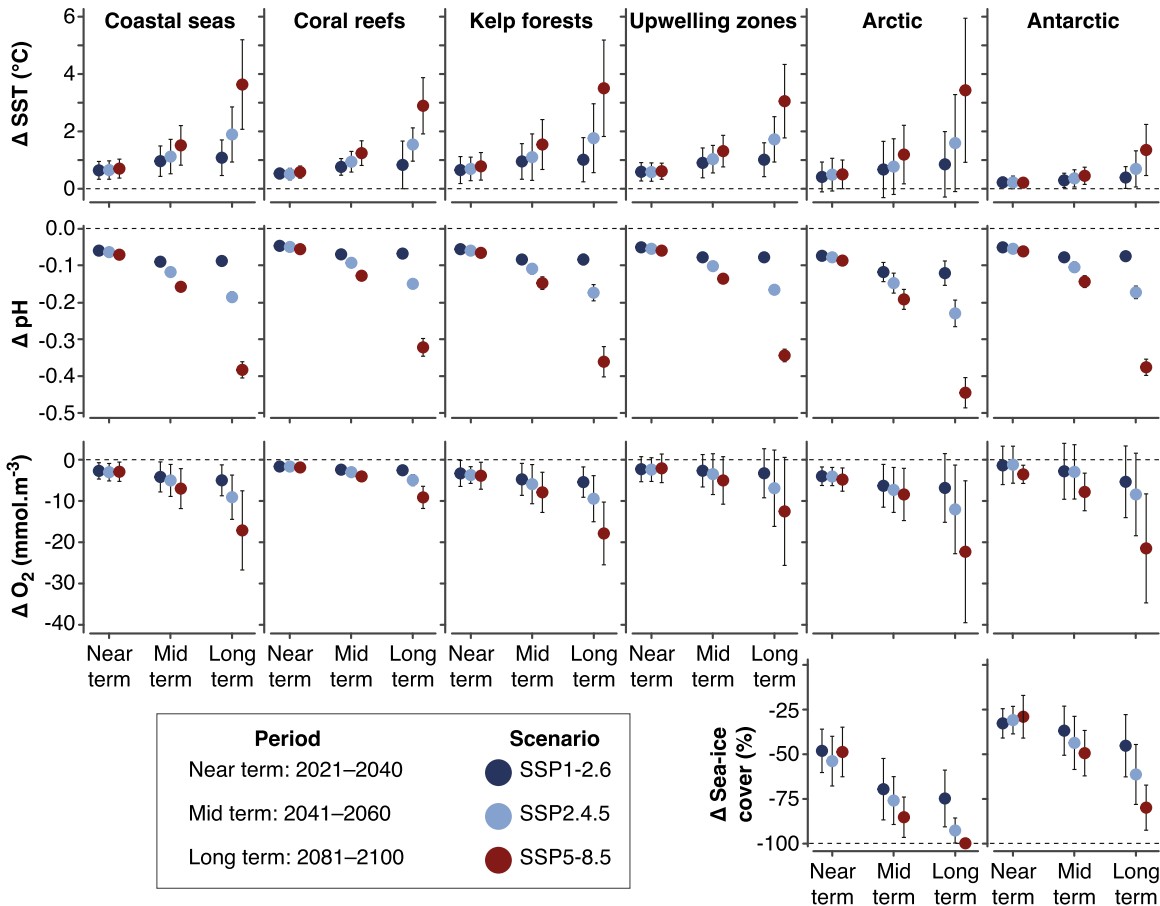

**Figure 2.** Projected changes in climate-induced drivers across coastal systems relative to the recent past (1985–2014), after Cooley et al. (2022). Climate-induced drivers are arranged by row, while coastal systems are arranged by column. All measures are for the ocean surface, except changes in oxygen concentrations, which are subsurface (100–600 m) in upwelling and polar systems. Projections are derived from an ensemble of CMIP6 models interpolated to a 1° x 1° grid. Error bars represent 90% confidence.

**Table 3.** Policy context of common future climate scenarios (Meinshausen et al., 2020; Chen et al., 2021). Scenarios are named by Shared Socioeconomic Pathway (SSPs) and radiative forcing level in 2100, approximating Representative Concentration Pathways (RCPs; W m$^{-2}$).

| Scenario | Policy relevance, including global warming levels (GWL) relative to the pre-industrial and corresponding 90% confidence intervals, as assessed by the IPCC Sixth Assessment Report (IPCC, 2021) |
|---|---|
| SSP1-1.9 | A 1.5°C world: an equitable world with sustainable development achieves net zero greenhouse-gas emissions by mid-century and maintains net negative emissions for several decades thereafter. This stabilises global temperatures at 1.4°C (1.0–1.8°C) GWL. Minimal overshoot beyond 1.5 °C is in line with the stretch goal of the Paris Agreement. |
| SSP1-2.6 | A 2°C world: an equitable world with sustainable development achieves net zero emissions around 2075 and maintains net negative emissions for several decades thereafter. Temperatures remain below the target set by the Paris Agreement, being restricted to 1.8 °C (1.3–2.4°C) GWL. |
| SSP2-4.5 | Approximates current climate policies: a world implementing current climate policies and following current trends of slow reduction in consumption and energy use, sees emissions rising until around 2050, before declining to net zero by 2100. Temperatures exceed the Paris Agreement, reaching 2.7° C (2.1–3.5°C) by 2100 and stabilising thereafter. Current nationally determined contributions (NDCs) for 2030 achieve a slightly lower GWL of 2.4°C (1.8–3.4°C) (Hausfather and Moore, 2022). |
| SSP3-7.0 | Approximates a scenario under which no new climate policy is implemented: a world retreating from globalisation to focus on domestic issues deprioritises the environment, resulting in slow economic growth and a doubling of emissions by 2100. Warming reaches 3.6°C (2.8–4.6°C) GWL by 2100 and continues thereafter. |
| SSP5-8.5 | An extreme counterfactual: a world focusing on capitalism, open markets and consumption results in rapid economic growth. Emissions double by 2050 and triple by the end of the century. Warming reaches 4.4°C (3.3–5.7°C) by 2100 and continues thereafter. |

related to ocean and coastal activities and users, developing new technologies and innovative built infrastructure, and employing marine and coastal nature-based solutions while also carefully managing existing and future coastal development (Duarte et al., 2020; Frazão Santos et al., 2020; Cooley et al., 2022; Shin et al., 2022; van Woesik et al., 2022; Pörtner et al., 2023; Rossbach et al., 2023).

Given this reality, together with the assessment that anthropogenic climate change has already exposed coastal ecosystems to conditions unprecedented over millennia, it can be projected with some confidence that the impacts already caused by climate change will become worse and more extensive (Cooley et al., 2022). Here, we present a selection of the most robust projections of future risk,

focusing on habitat-forming species due to their importance in ecological structure and functioning.

## Projected climate risks for habitat-forming coastal taxa

Evidence from the palaeorecord and from species-environment relationships suggests that ocean warming and acidification will result in declining coral reef extent and species richness (Pandolfi et al., 2011; Pandolfi and Kiessling, 2014; Hoegh-Guldberg et al., 2018a; Hoegh-Guldberg et al., 2019; van der Zande et al., 2020; Chaudhary et al., 2023). Recent assessments that exclude consideration of natural adaptive capacity project declines in reef extent by >70% at 1.5°C of warming, and by >99% at 2°C (Hoegh-Guldberg et al., 2018b; Kalmus et al., 2022). Considering return times of severe bleaching-level heat events provides an even more pessimistic outlook (Kalmus et al., 2022). However, on the basis of results from a coral-symbiont eco-evolutionary model, Logan et al. (2021) concluded that natural adaptation – including evolution and other processes – could allow 70–80% of coral to survive the century with ~2°C of warming, but with warming beyond 3.5°C by 2100, <10% of coral cover would remain. The relatively high levels of trait heritability among corals (Bairos-Novak et al., 2021) further emphasise the potential for future adaptation. However, limits remain, and resilience is spatially variable, even when accounting for adaptive capacity (Cornwall et al., 2023).

Kelps and seagrasses are also at risk from future warming (Cooley et al., 2022). For both taxa, warming trends commonly underlie projections of extirpations at warm range edges, with poleward range extensions at cool range edges (Wilson and Lotze, 2019; Assis et al., 2022; Davis et al., 2022; Pecquet et al., 2022; Daru and Rock, 2023). However, there are exceptions to this pattern (e.g., Goldsmit et al., 2021), especially for invasive seagrasses (Wesselmann et al., 2021). Some species might even find refuge from warming at depth and boost their overall biomass due to increasing productivity in the cooler parts of their ranges (Davis et al., 2022). The potential effects of projected changes in marine heatwaves remain qualitative (e.g., Starko et al., 2022), although progress is being made towards more quantitative projections (e.g., Pruckner et al., 2022; Li and Donner, 2023).

Other climate-sensitive coastal taxa that contribute to habitat structure include mangroves and saltmarshes. But even for the well-studied mangrove forests, quantitative projections are challenged by difficulties in disentangling the impacts of recent climate change from those of other non-climate anthropogenic stressors, and by the mix of positive (e.g., through enhanced productivity due to $CO_2$ enrichment and warming) and negative (e.g., due to sea-level rise and drought) effects of projected climate change (Friess et al., 2022). Further complications are introduced by uncertainties surrounding future trajectories of socioeconomic development in the coastal zone, which can have effects on projected gains or losses of coastal wetland habitat that at least equal those of climate change (Ouyang et al., 2022; Liang et al., 2023). Central to the future of these systems is the availability of accommodation space, which regulates their ability to accrete sediment and move inland in response to sea-level rise (Krauss, 2021; Rogers, 2021). In this sense, coastal development that restricts accommodation space and traps wetlands in a coastal squeeze is arguably the largest threat to their resilience (Cooley et al., 2022). Irrespective, analysis of reconstructed palaeorecords suggests projected rates of sea-level rise (Table 1) will overwhelm the ability of mangroves to keep pace with rising water levels by mid-century, even with ambitious mitigation of greenhouse-gas emissions, and that saltmarshes face the same fate by the end of

the century (Horton et al., 2018; Saintilan et al., 2020; Törnqvist et al., 2020). Although the vulnerability of coastal wetlands decreases with increasing sediment availability, greater elevation on the shore, and increasing tidal range (Schuerch et al., 2018; Saintilan et al., 2020; Friess et al., 2022), efforts at restoration and protection appear to be the key to the future resilience of these systems. Quantitative projections remain contentious for other coastal systems, such as sandy beaches (e.g., Cooper et al., 2020; Vousdoukas et al., 2020a; Vousdoukas et al., 2020b).

## Broader projections of climate risks for marine taxa and regions

When considering quantitative projections of processes such as range shifts (e.g., García Molinos et al., 2016; Gokturk et al., 2022) and phenology (e.g., Asch et al., 2019; Gokturk et al., 2022; Yamaguchi et al., 2022), analyses are often either taxon-specific or global, and not focused specifically on coastal ecosystems. The same is true for quantitative projections of future risk. Yet results are instructive for coastal ecosystems. For example, Trisos et al. (2020) project not only that temperatures across entire species' ranges will transition to levels unprecedented in those species' recent (1850–2005) experience, but that this will occur abruptly, especially for marine species such as seagrasses, corals, cephalopods, marine reptiles and marine mammals. This phenomenon is projected to manifest in tropical oceans before 2030 under the highest emissions scenarios and escalate with duration and the magnitude of emissions. Pigot et al. (2023) confirm this trend towards abrupt thermal exposure across species' ranges. Further evidence comes from a recent analysis of future climate risk across ~25,000 marine species (Boyce et al., 2022), which found that by the end of the century, risk was substantially reduced for ~1.8°C relative to ~4.4°C of global warming, with 1.3% vs 2.7% of assessed species being at critical risk and 54% vs 84% at high risk, respectively. Since many of the taxa in these analyses occur primarily in coastal waters, these generic risks may be assumed to hold there, too.

Moving beyond projections of climate risk for individual taxa becomes more difficult, but climate analogues can help. For example, on the basis of data from the palaeorecord, Reddin et al. (2022) project that if warming levels approach those anticipated under the highest emissions scenarios (Table 3), taxa with thermal optima beyond ~21°C will experience elevated risk of extinction, as will those with thermal optima below ~11°C. This pattern is mirrored by projections from an ecophysiological model validated on spatial patterns of extinction from the fossil record, which projects extirpations at the tropics and extinctions at the poles, but with substantial reductions in risk from immediate and strong mitigation (Penn and Deutsch, 2022). Again, however, these are generic projections, and coastal taxa must be assumed to comply with reported patterns.

## The emergence of ecosystem and global models for projecting of climate risks in the ocean

The growing need for policy advice in the face of these projected climate-change risks has challenged the scientific community to develop models that go beyond exploring risks to biodiversity, such as those discussed above, to instead assess risks to ecosystem functioning and service provision (Weiskopf et al., 2022). This requires modellers to build from familiar outputs of Earth System models (ESMs), such as changes in temperature, pH and salinity, and, more recently, nutrient availability, phytoplankton and even

zooplankton biomass (Canadell et al., 2021), to project changes in primary and secondary production.

One response has been the development of the Fisheries and Marine Ecosystem Model Intercomparison Project (FishMIP), which specifies sets of common ESM forcings and model outputs for a diverse suite of marine fisheries and ecosystem models (Tittensor et al., 2018). The advantage of the "modelling intercomparison" approach is that it yields comparable projections from each model, allowing assessment of the range of plausible outcomes, given our current understanding and computing capacity (Heymans et al., 2020). This is important because marine ecosystem models still largely lack formal approaches to validation, calibration, and quantification of uncertainty (Steenbeek et al., 2021).

In terms of uncertainty, scenario uncertainty can be assessed by comparing model outputs generated under different future scenarios, such as those in Table 3. But parametric uncertainty – the uncertainty around the parameters within individual models – is far more difficult to address for models with any level of complexity, given current computational capacity (Steenbeek et al., 2021). Finally, intercomparison can help to quantify structural – inter-model – uncertainty, but attempts to do so reveal that the projected responses to two of the most fundamental inputs to the models – magnitude of warming and productivity of lower trophic levels – are inconsistent in both direction and magnitude amongst models (Heneghan et al., 2021). This disparity emphasises that understanding of how ecosystem-level effects emerge from individual-level processes remains incomplete. Despite limitations in individual ecosystem models, ensembles of models – such as those in FishMIP – provide projections of global decline in total biomass of marine animals that worsen with increasing emissions (Tittensor et al., 2021), with reasonable agreement in the direction of change in coastal systems, but little agreement on the magnitude of change (Cooley et al., 2022).

## A view of the way ahead

Despite the recent advances in our understanding of the impacts of anthropogenic climate change on coastal ecosystems described in Cooley et al. (2022) and updated above, projections of future climate risk have not progressed much beyond the notion that risks escalate with cumulative emissions (i.e., with warming). Part of the problem is that our detailed understanding of processes at the single-organism level seldom adequately addresses interactions among multiple drivers, so does not scale intuitively to predict integrated responses at the levels of populations or ecosystems (Boyd et al., 2018; Collins et al., 2022). Theoretical progress is being made in this regard (Orr et al., 2020; Pirotta et al., 2022), but proposed solutions are not yet commonly implemented. This problem is exacerbated for coastal ecosystems by the interactions among numerous climate-induced and non-climate anthropogenic drivers (Table 2; Gissi et al., 2021). Not only do these interactions complicate the parameterisation of models, but they also complicate the scenario space that must be explored: what humans do in the coastal zone can often have a larger ecological effect than that of climate change. For example, under ambitious mitigation, conservation and coastal-zone planning can ameliorate climate impacts on coastal ecosystems, but climate-uninformed coastal development can condemn those same ecosystems (Cooley et al., 2022). The utility of advice to policymakers therefore requires

more purposeful selection and articulation of scenarios to be considered – simply making projections for SSP1-2.6 and SSP5-8.5 cannot remain the norm. In this context, one solution might be to consider warming levels (e.g., 1.5°C, 2°C and 3°C relative to pre-industrial), each in combination with alternative coastal development scenarios. This is impractical with current CMIP6 model outputs because each scenario subsumes a shared socioeconomic pathway (Table 3), but the increasingly prominent use of climate-model emulators (Nicholls et al., 2020; IPCC, 2021) might provide a solution.

There are several other areas where progress is urgently needed. But one requires particular attention because we have not addressed it elsewhere in this review: the need to expand the knowledge systems on which our assessments depend by ensuring that Indigenous Knowledge and perspectives are more adequately represented in our assessments (Fischer et al., 2022; Schipper et al., 2022). Silent cores of sediment and coral have taught us so much about the past and the future; how much more could we learn from the rich oral histories held by Indigenous Peoples around the world?

**Open peer review.** To view the open peer review materials for this article, please visit http://doi.org/10.1017/cft.2023.27.

**Acknowledgements.** We acknowledge the contributions of Laurent Bopp, Philip Boyd, Simon Donner, Shin-Ichi Ito, Wolfgang Kiessling, Paulina Martinetto, Elena Ojea, Marie-Fanny Racault, Björn Rost, Mette Skern-Mauritzen, Dawit Yemane Ghebrehiwet, Lisa Levin and Karim Hilmi, who were Lead Authors or Review Editors on Chapter 3 of the IPCC Sixth Assessment Report: Oceans and Coastal Ecosystems and Their Services. Without their work, neither that assessment nor this paper could have been completed. We also acknowledge the many Contributing Authors and Expert Reviewers for the IPCC Chapter, especially Olivier Torres, who provided the data for Figure 2.

**Author contribution.** All authors contributed to all aspects of the development of this manuscript.

**Financial support.** D.S.S. was funded by Australian Research Council Discovery Project DP230102359. J.A.B. was funded by an Australian Government Research Training Program Scholarship and a Commonwealth Scientific and Industrial Research Organisation ResearchPlus Scholarship.

**Competing interest.** The authors declare no competing interests exist.

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
