## [Reviewer Report]

Shoeman et al. review climate change impacts on coastal marine ecosystems, seemingly refining and updating parts of Chapter 3 of the IPCC AR6 WGII report. The manuscript here is wide-ranging and timely, summarizing key information across a range of ecosystems, drivers, approaches, and outcomes. In particular, the authors describe and recognize the substantial challenges, particularly around detection, attribution and scenario usage, and do a good job of integrating the broader marine literature when coastal-specific information is lacking. The work is well organized and provides a useful compilation that will have a broad appeal to the scientific community. I provide some suggestions and comments for consideration below.

● The detection and attribution framework is well explained, but I do not find Figure 1 particularly clear. The schematic representations for each aspect of the framework are quite cluttered and it is difficult to pick out the key points and implied flow. I appreciate that, for example, it is difficult to find an image that represents ‘mechanism for impact’, but this suggests that perhaps the whole approach needs a rethink as a diagram should be intuitive and aid understanding. I am also not sure that the plus and equals signs make sense or add anything in this context; they just confuse things and it may be better to use a number for each step (e.g. 2. Vulnerability) to indicate the sequential process. I suggest that the authors consider other approaches (for example, a table may be a clearer way to present these steps, and perhaps they could include a simple icon in each row of the table to add visual appeal). I also suggest that the authors look at Gonzalez et al. (2023; Phil. Trans. R. Soc. B) which explores detection and attribution for biodiversity, and has an alternative diagram for a similar workflow that I find cleaner and easier to interpret.

● The discussion on scenarios is both interesting and important. However, does it reflect the debate within the scientific community about this topic? For example, the Schwalm et al. (2020, PNAS) response to Hausfather & Peters (2020) suggests that cumulative CO2 emissions are consistent with RCP8.5 and that it remains a useful scenario. There is also the potential for climate tipping points beyond 1.5C (e.g. see McKay et al. 2022, Science) - could this not affect the plausibility of different scenarios? SSP1-2.6 may still be attainable, but is it the case that it relies upon large-scale deployment of as-yet unproven technology? If so, how likely is it that we achieve these breakthroughs? It would be tremendously helpful if the authors could help steer those (such as myself) who are following these debates through the complexities of this topic by providing more information and guidance on these issues.

● The authors do not discuss how changes in ocean circulation patterns may affect marine ecosystem (as distinct from upwelling or stratification); if this is due to large uncertainties, then I would still recommend mentioning potential impacts in the ‘other climate-induced drivers’ section.

● The phrasing around ‘secular trends’ may be less familiar to marine ecologists, so I suggest providing a definition.

● Line 160-161: Just a caution about the wording; there can be dramatic spatial gradients in temperature in the vertical axis.

● Lines 208-210: Although the authors are discussing marine heatwaves rather than thermally driven range-shifts, it still seems a bit risky to indicate that effects can be beneficial and use reduced interactions between whales and pot-fishing gear as an example. The authors give a citation that suggests that this is true, and one that doesn’t, but also see Record et al. 2019, which suggests that climate-driven changes have lead to greater vulnerability of NA right whales to gear entanglement. I would suggest listing effects as beneficial only when the evidence is near unequivocal.

● The section on compound events modifying ecological interactions only has a very short discussion of metabolic activity, focussing on impacts on parasites and health. I would suggest that the authors explore other potential ecological impacts of temperature-driven metabolic cost changes, such as on growth, energetic allocation, potential predation interactions (e.g. see Grady et al. 2019 Science), fecundity etc – or at least indicate that the potential impacts are more broad than those currently explored in the paper. (Note also that the Heneghan et al. 2021 paper that is cited in the manuscript also suggests that metabolic consequences may play an important role in explaining the differing responses of marine ecosystem models to climate change).

● Lines 366-367: I’m not sure about the wording here. Many terrestrial systems are also heavily impacted by non-climate anthropogenic drivers and are also vulnerable to the coupled climate and biodiversity crises. Suggest deleting the ‘uniquely’ or otherwise clarifying.

● Lines 383-389: The section on climate-smart conservation planning is somewhat reductive, essentially boiling down to ‘reduce other anthropogenic stressors’. This doesn’t reflect the broader array of operational, management, spatial planning, and policy tools that may be needed. The wording around reducing anthropogenic stressors in areas where ecosystems are least exposed to projected climate change also doesn’t reflect the nuances of the situation – reducing stressors may be equally important in areas that are undergoing high levels of climate change due to the greater imposed stress on ecosystems and organisms therein. See Tittensor et al. (2019, Science Advances) for more on these issues. (Lines 444-449 also follow a similar thread).

● Line 398: Suggest ‘statistical’ rather than ‘numerical’

● Line 408: Whilst I like the idea of ecological models as predators, I think ‘predictors’ is meant here. (Actually, projecting rather than predicting may be better).

● Line 411 onwards: I am glad that the section on paleoecology is in here; it is indeed an approach that can aid in our understanding. I further suggest looking into the work of Moriaki Yasuhara who has done a lot of exploration in this space.

● Mangroves, kelp, and coral are discussed in some detail, but not seagrasses. I suggest adding a little more.

● Line 554: This is important, and very true, but note that the new ISI-MIP 3a simulation round (of which FishMIP is a part) is centred around attribution (Frieler et al. 2023, EGUsphere), so this should improve.

In summary, this is a polished, wide-ranging and useful review that condenses a lot of useful information across multiple scientific research areas, and provides guidance around next steps for our broader shared research efforts.

---

## [Reviewer Report]

A nice overview - I have some specific comments:

line 49: this IPCC report - are you referring here to literature published since the cut off date for the IPCC WGII AR6?

Line 54 and line 60: specify greenhouse has emissions

Line 86 - as Table 1 does not give unprecendented examples, suggest making this a separate sentence and summarise the examples from below

Line 102-111: what about OMZs?

Line 140-151: worth highlighting the role of multiple lines of evidence here see WGII Chp 1 box Attribution

Line 188: tropicalisation is used specifically for arrival of tropical species into regions, at higher latitudes similar processes occur eg borealisation in Arctic - this needs clarification

Line 285: remove acronym

Line 395: hazard used elsewhere

Lines 420-432: it may be worth bringing in the scenario followed will depend on ambition and action eg https://www.science.org/doi/full/10.1126/science.adg6248

---

## [Editor Report]

Overall, I think this manuscript is well written and only needs minor revisions. A few comments - Figure 1 is confusing and I recommend a revision to make the point clearer. Reviewer 1 also provided feedback. Can you address how other sea grasses may be impacted beyond kelp? Line 128 is the only header that is posed as a question. Please consider editing to match the rest of the paper.